



# Increased vertical resolution in the stratosphere reveals role of gravity waves after sudden stratospheric warmings

Wolfgang Wicker[1], Inna Polichtchouk[2], and Daniela I. V. Domeisen[1,3]

[1]Université de Lausanne, Lausanne, Switzerland
[2]European Centre for Medium-Range Weather Forecasts, Reading, UK
[3]ETH Zurich, Zurich, Switzerland

**Correspondence:** Wolfgang Wicker (wolfgang.wicker@unil.ch)

**Abstract.** Sudden stratospheric warmings (SSW) have a long-lasting effect within the stratosphere as well as impacts on the underlying troposphere. However, sub-seasonal forecasts of the winter polar stratosphere fail to use their full potential for predictability as they tend to underestimate the magnitude and persistence of these events already within the stratosphere. The origin of this underestimation is unknown. Here, we demonstrate that the state-dependent stratospheric cold bias following

SSW events in sub-seasonal hindcasts can be halved by increasing vertical model resolution, suggesting a potential sensitivity to gravity wave forcing. While the predictability of the planetary Rossby wave flux into the stratosphere at lead times longer than a week is limited, the existence of a critical layer for gravity waves with a small zonal phase speed caused by the disturbed polar vortex provides predictability to the upper stratosphere. Gravity wave breaking near that critical layer can, therefore, decelerate the zonal flow consistently with anomalous subsidence over the polar cap leading to warmer temperatures in the

middle polar stratosphere. Since the spectrum of gravity waves involves vertical wavelengths of less than 4000 m, as estimated by wavelet analysis, a high vertical model resolution is needed to resolve the positive feedback between gravity wave forcing and the state of the polar vortex. Specifically, we find that at a spectral resolution of TCo639 (approximate horizontal grid-spacing of 18 km) at least 198 levels are needed to correctly resolve the spectrum of gravity waves in the ECMWF Integrated Forecasting System. Increasing vertical resolution in operational forecasts will help to mitigate stratospheric temperature biases

and improve sub-seasonal predictions of the stratospheric polar vortex.

## 1   Introduction

Atmospheric waves entering the extra-tropical stratosphere from below comprise a wide range of spatial scales. The large-scale part of the spectrum is occupied by Rossby waves with zonal wavenumbers 1-3, often termed planetary waves. Depending on their phase speed and the zonal-mean zonal background wind, planetary waves can propagate from the troposphere into the

stratosphere where they break, deposit easterly momentum, and slow down the polar vortex (e.g. Charney and Drazin, 1961; Domeisen et al., 2018). A particularly strong and sustained planetary wave flux can lead to a complete breakdown of the polar vortex, a reversal of the zonal-mean westerlies, and a rapid warming of the polar stratosphere which is termed a sudden stratospheric warming (SSW) (Matsuno, 1971; Baldwin et al., 2021). While some SSWs are associated with Rossby wave





reflection and have no long-term impact (Kodera et al., 2016), the majority of events decay slowly over the course of a month
or longer (Limpasuvan et al., 2004).

Gravity waves with horizontal wavelengths from ∼10 to ∼1000 km and frequencies between the Brunt-Väisälä frequency $N$ and the inertial frequency $f$ form the small-scale part of the spectrum (e.g. Fritts and Alexander, 2003). Typically, extra-tropical gravity waves are excited near the surface by flow over orography or in the upper troposphere by jet/front imbalances. These waves propagate via the stratosphere into the mesosphere, where their amplitudes grow until they break. However,
depending on their phase speed and the background wind, gravity waves can encounter a critical layer, where vertical length scale shrinks to zero, and deposit their momentum already at lower altitudes in the stratosphere. For medium-frequency waves with frequencies $\omega \gg f$ the critical layer is often defined as the surface where the zonal-mean zonal wind equals the phase speed, whereas low-frequency waves meet their critical layer when their intrinsic frequency $\hat{\omega}$ approaches the inertial frequency $f$ (Fritts and Alexander, 2003). Compared to planetary waves, gravity waves receive less attention in extra-tropical stratosphere
studies. However, both observational and modeling studies document increased gravity wave amplitudes at the edge of the polar vortex during minor stratospheric warmings concurrently with the peak of the planetary wave flux (e.g. Duck et al., 1998; Venkat Ratnam et al., 2004; Wang and Alexander, 2009; Yamashita et al., 2010; Dörnbrack et al., 2018; Polichtchouk and Scott, 2020). During major warmings, the polar vortex breaks down and the downward propagation of the zero-wind line prevents the propagation of stationary gravity waves into the upper stratosphere and mesosphere (e.g Wang and Alexander,
2009; Hindley et al., 2020). The absence of gravity wave breaking in the mesosphere explains the mesospheric cooling during an SSW by a relaxation to radiative equilibrium (Holton, 1983). Moreover, during Southern Hemisphere spring-time polar vortex breakdown, gravity waves are found to contribute significantly to the polar vortex deceleration in ERA5 reanalysis (Gupta et al., 2021). While recent studies consider the role of gravity waves in pre-conditioning the vortex for planetary waves before the onset of vortex split events (Albers and Birner, 2014; Song et al., 2020; Kuchar et al., 2022), the present study
focuses on the effect of gravity waves on the zonal-mean zonal momentum after the central date of an SSW.

Due to their downward response (Baldwin and Dunkerton, 2001), SSWs hold great potential for sub-seasonal to seasonal prediction (Domeisen et al., 2020a), and an accurate simulation and representation of the winter stratosphere in weather and climate models is important for understanding and predicting tropospheric impacts of stratospheric variability. While sub-seasonal forecasts of the winter stratosphere are often skillful (e.g. Tripathi et al., 2015a; Domeisen et al., 2020b), state-of-the-
art prediction models are still biased. Specifically, it has been found that the magnitude of SSWs is generally underestimated (e.g. Karpechko et al., 2018; Lawrence et al., 2022). Another source of tropospheric errors for sub-seasonal forecasts can be a misrepresentation of stratosphere-troposphere coupling and the downward influence of SSWs (Kolstad et al., 2020). To our knowledge, these forecasting errors have not been connected to a potential underestimation of resolved gravity wave momentum fluxes. One possible model adjustment to improve sub-seasonal prediction is an increase in the vertical resolution.
Generally, vertical resolution receives less attention than horizontal resolution in atmospheric modeling studies but it is well established that vertical resolution needs to be set in consideration of its horizontal counterpart (e.g. Lindzen and Fox-Rabinovitz, 1989; Roeckner et al., 2006; Skamarock et al., 2019). The representation of Rossby waves requires a consistent





aspect ratio of the vertical and horizontal grid spacing $\Delta z/\Delta L$ determined by the ratio of the scale height to the radius of deformation (Lindzen and Fox-Rabinovitz, 1989)

$$\frac{\Delta z}{\Delta L} \approx \frac{f}{N} \tag{1}$$

To resolve the regime of stratified turbulence at small horizontal scales, the vertical grid spacing depends primarily on the stratification and can be set following the ratio

$$\Delta z \approx \frac{U}{N} \tag{2}$$

of the horizontal velocity scale $U$ to the buoyancy frequency $N$ (Waite, 2016; Cullen, 2017). For the undisturbed polar vortex, this gives a vertical grid spacing of $\mathcal{O}(1\mathrm{km})$ in the stratosphere. For a disrupted vortex with $U = 0\,\mathrm{m\,s^{-1}}$, on the other hand, this relation would require infinite vertical resolution. The vertical wavelength of a gravity wave approaching its critical layer shrinks to zero. According to Lindzen and Fox-Rabinovitz (1989), a requirement for infinite vertical resolution is only prevented by the presence of a complex phase speed, i.e. damping. Approximating the imaginary part of the phase speed using a linear damping rate $\sigma_i$, they postulate an aspect ratio of the vertical to the horizontal grid spacing

$$\frac{\Delta z}{\Delta L} \approx \frac{\sigma_i}{N} \tag{3}$$

However, the damping rate $\sigma_i$ can be very difficult to infer and dedicated models experiments are required to determine the correct settings of vertical resolution at a given horizontal resolution, as we do in the present study. Given the strong constraint for vertical resolution near a critical layer and the widely different horizontal scales (Lindzen and Fox-Rabinovitz, 1989), higher sensitivity can be expected for processes involving gravity waves than for planetary waves.

A detailed description of the model experiments is given in Section 2. In Section 3 we confirm a vortex state-dependent temperature bias, i.e. an underestimated SSW amplitude, in hindcasts from the S2S database and demonstrate how this bias can be mitigated targeted model simulations by increasing vertical resolution. The benefit of vertical resolution for gravity wave drag and potential energy is presented in Section 4. A discussion of these results follows in Section 5.

## 2 Data and methods

For this study, we conduct model experiments with the ECMWF Integrated Forecasting System (IFS CY47R3) with different vertical resolutions. The subject of these experiments is the simulation of the major SSW in boreal winter 2018 and its downward influence on the tropospheric circulation on sub-seasonal timescales. Most of these hindcasts are initialized on 8th February and run for 46 days. For robustness analysis, additional simulations are initialized on 17th January 2006 and 5th February 2010 that confirm our findings for a vortex displacement and another split SSW event as discussed in Appendix A. For





85    each start date and each model configuration, we initialize an ensemble of 51 realizations. The output of individual realizations is used whenever we compute quadratic quantities such as variance, eddy-heat, or eddy-momentum fluxes. Ensemble-mean anomalies from climatology are regarded as the predictable signal in the sense of probabilistic predictability as is commonly done for sub-seasonal prediction. Initial conditions are taken from the ERA5 reanalysis dataset (Hersbach et al., 2020), which is also the reference against which we evaluate the model experiments.

90    The bulk of our simulations is run at a spectral horizontal resolution of TCo639, which corresponds approximately to 18 km grid-spacing, and a vertical resolution of either 91 or 198 levels in hybrid $\eta$-coordinates with a sponge layer starting at 1 hPa. The vertical grid-spacing for the different model configurations is illustrated in Figure 1 depicting high vertical resolution near the surface and a gradually coarsening vertical grid in the stratosphere. Sensitivity tests for a comparison with operational forecasts at ECMWF reveal that while 137 vertical levels seem sufficient at a spectral resolution of TCo319 (36 95    km grid-spacing), at least 198 vertical levels are required at TCo639 (see Section 5 and Appendix B).

Changing resolution will presumably affect the magnitude of resolved gravity wave momentum flux. However, at a horizontal resolution of approximately 18 km the stratospheric gravity wave spectrum is only partly resolved and sub-grid scale gravity waves need to be parameterized. Note that a comparison of ECMWF IFS simulations suggest that the gravity wave spectrum is not fully resolved even at a 4 km horizontal grid-spacing (Polichtchouk et al., 2022). The settings for parameterized gravity 100    wave drag are unchanged between the different vertical resolution configurations. To verify whether changes in gravity wave drag are caused directly by changes in the vertical resolution as opposed to changes in the stratospheric zonal-mean basic state, we performed nudged simulations following the SNAPSI protocol (Hitchcock et al., 2022). Specifically, zonal-mean temperature, vorticity, and divergence above 90 hPa are relaxed towards the 'observed' time series of ERA5 on a time-scale of 6h. Consequently, there are no significant differences in the zonal-mean zonal wind and polar-cap temperature anomaly 105    between the different model configurations in the nudged simulations.

The vertical eddy-momentum flux convergence on pressure levels $\left[-\partial \overline{u'\omega'} / \partial p\right]$, where $u$ and $\omega$ are zonal and vertical (pressure) velocities and $p$ is pressure, is used as the diagnostic for the zonal resolved gravity wave forcing. We follow the methodology of Polichtchouk et al. (2022) and represent primes as all spherical harmonics with total wavenumbers between 21 and the truncation limit. Selecting only these harmonics explicitly eliminates the contributions by planetary and synoptic-110    scale waves. The very same horizontal filtering is used when diagnosing the vertical power spectra of potential temperature anomalies as it is explained in the following. The bar denotes an average over the largest resolved wavelength.

In order to estimate vertical wavenumbers $m$ and vertical wavelengths $2\pi/m$ we use log-pressure coordinates where the following wave ansatz can be made (Fritts and Alexander, 2003)

$$\frac{\theta'}{\overline{\theta}} = \widetilde{\theta} \cdot \exp\left[imz + \frac{z}{2H}\right] \tag{4}$$

The scale height $H$ is set to 7000 m, $\theta$ is potential temperature, and $\widetilde{\theta} = \widetilde{\theta}(x,y,t)$ is the amplitude of a normal mode at a point in $(x,y,t)$. To fully use the vertical resolution of the model, we interpolate potential temperature data from native model levels to a constant grid spacing of 100 m (marked by the vertical line in Fig. 1) instead of using standard pressure level output. Then



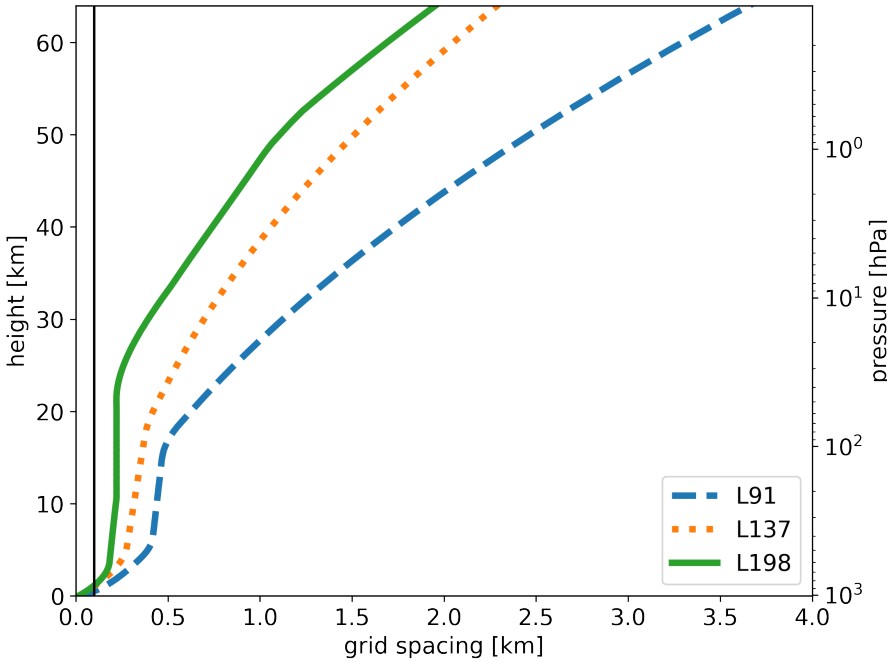

**Figure 1.** Vertical grid spacing in log-pressure coordinates with a scale height of 7000 m for three vertical grid configurations with standard surface pressure. The vertical line marks a constant grid spacing of 100 m.

we apply a wavelet analysis, i.e. convolve profiles of normalized anomalies $\left[\exp\left(-z/2H\right)\left(\theta'/\overline{\theta}\right)\right]$ with a Morlet wavelet (e.g. Torrence and Compo, 1998) to compute spectral power and estimate the dominant vertical wavenumber at each altitude. The

spectral power of $\left[\exp\left(-z/2H\right)\left(\theta'/\overline{\theta}\right)\right]$ is proportional to potential energy per unit volume. To remove artifacts introduced by the lower boundary of the model domain, we apply a Hanning window tapering to the vertical column of normalized perturbations.

To confirm the state dependence of stratospheric temperature biases on the strength of the polar vortex and to demonstrate the relevance of our simulations, we analyze the extended winter (NDJFM) IFS hindcasts provided by ECMWF to the S2S

database (Vitart et al., 2017). Specifically, we estimate composites of polar cap temperature biases at 50 hPa using zonal-mean zonal wind at 10 hPa and 60 °N in reanalysis during initialization as the criterion to select initialization dates and construct the composites (Tripathi et al., 2015b). The thresholds of 40 m s$^{-1}$ for a strong vortex state and 5 m s$^{-1}$ for a weak vortex state are chosen following Domeisen et al. (2020a) to ensure sufficiently large sample sizes. This sample of S2S hindcasts comprises more than 21,000 ensembles members for the winters 1999-2020 including 4191 members initialized during strong vortex

states and 2761 members during weak vortex states. The significance of selecting the composite based on the strength of the vortex during initialization is tested using a Monte Carlo technique by comparing the observed difference with the distribution of hindcast biases for randomly selected start dates.





## 3  Increased vertical resolution strengthens SSW temperature anomaly in the polar stratosphere

Over the polar cap, the ECMWF S2S hindcasts develop a mid-to-lower stratospheric cold bias of more than 2 K at 50hPa with
increasing lead time (dashed grey line in Fig. 2a), which is in line with the findings of Lawrence et al. (2022) for high-top
models. Hindcasts with weak vortex initial conditions develop a stronger than average cold bias (dotted blue line in Fig. 2a).
Specifically, the difference between the weak vortex composite mean and the climatological bias reaches its maximum of about
-0.5 K after 20 days into the forecast before the influence of the initial conditions shrinks and an increasing number of composite
members reach a neutral or strong vortex state (Fig. 2b). This aggravated cold bias corresponds to an underestimated magnitude
of weak vortex events. Similarly, Lawrence et al. (2022) find that at a lead time of one week, the wind change associated with
an SSW is underestimated by approximately $5 \, \mathrm{m \, s^{-1}}$. In contrast to weak vortex events, the strong vortex composite mean is
not significantly different from the climatological bias for the first two weeks of the hindcast until horizontal model resolution
is reduced and the temperature bias diminishes by 0.25 K compared to the average (Fig. 2, see Appendix B for a discussion of
the model resolution in S2S hindcasts). A comparison with the Monte Carlo distribution in Figure 2b shows that the observed
difference lies outside of the 99.9% significance which confirms that the null hypothesis can be rejected and that the cold bias
is significantly aggravated following weak vortex conditions. The existence of a critical layer in the upper stratosphere during
weak vortex states and underestimated resolved gravity wave forcing due to insufficient vertical resolution offer one potential
explanation for the vortex state dependant bias in the S2S hindcasts.

The analysis of a case study for the 2018 boreal winter SSW can be fruitful for a better understanding of the mechanisms
that might cause biases in the large ensemble of hindcasts. The central date of an SSW in reanalysis is typically preceded
and followed by a downward propagation of positive polar cap temperature anomalies on the order of 15 K to 20 K from the
stratopause to the lower stratosphere, as shown in Figure 3a for the 2018 event. The recovery of the polar vortex is marked
by the downward propagation of negative temperature anomalies from the mesosphere to the mid-stratosphere in the weeks
following the event. In the lower stratosphere, on the other hand, positive anomalies of about 5 K are sustained for six weeks
after the 2018 event. While the general development of the SSW is reproduced in sub-seasonal hindcasts, the magnitude of
sustained positive temperature anomalies is underestimated by 2 K to 4 K (Fig. 3b). Increasing vertical resolution from 91
levels to 198 levels reduces this negative bias significantly and prolongs the stratospheric warm temperature anomalies in the
ensemble mean (Fig. 3c). By comparing these results with simulations of SSW events in 2006 and 2010 (Fig. A1, A2) it is
confirmed that the strengthening and the extension of the stratospheric warming signal in time by increased vertical resolution is
robust amongst different SSW events. While the stratospheric signal amongst different SSW events shows strong resemblance,
the timing and character of the downward response can be very different. So, despite the importance of the stratospheric signal
for tropospheric predictability following an SSW (e.g. Domeisen et al., 2020a; Kautz* et al., 2020), no robust changes are seen
at the surface.

An evident hypothesis to explain the reduction of the cold bias might be that the planetary wave flux into the stratosphere
is improved by increased vertical resolution. The meridional eddy-heat flux at $100 \, \mathrm{hPa}$ shown in Figure 3d is used as a
proxy for the vertical Rossby wave flux in the lower stratosphere. Between Feb 08 and Feb 12, shortly after initialization

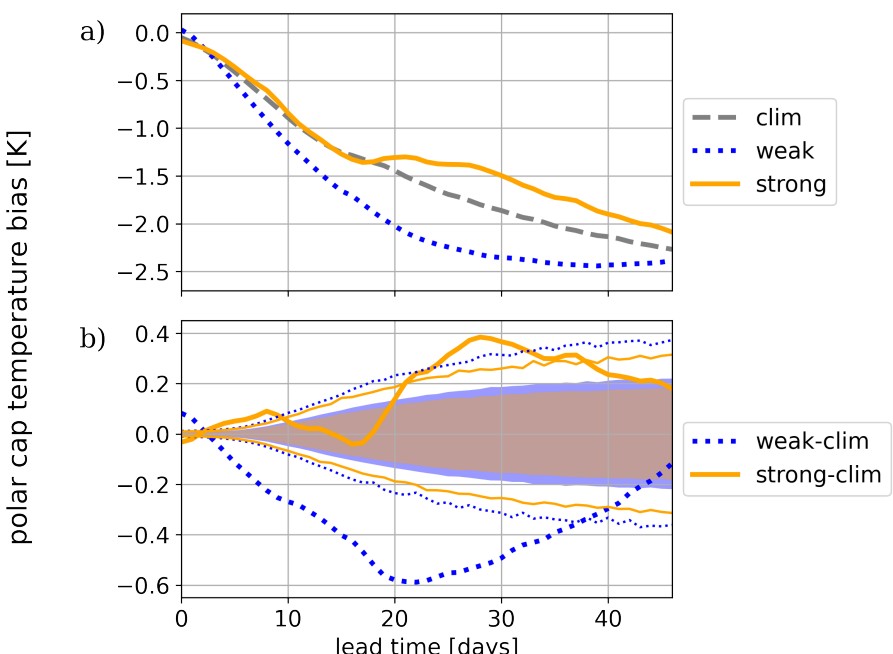

**Figure 2.** Polar cap temperature bias (hindcasts - reanalysis) at 50 hPa horizontally averaged between 60 °N and 90 °N of ECMWF S2S hindcasts plotted against lead time. Panel a) shows as a dashed gray line the average over all hindcasts ("clim"), as a dotted blue line the composite mean of hindcasts initialized during a weak vortex state ("weak"), and as a solid orange line the composite mean of hindcasts initialized during a strong vortex state ("strong"). Panel b) compares the difference between the "weak" composite bias (thick dashed blue) or the "strong" composite bias (thick solid orange) and the climatological bias with the 2.5% and 97.5% (shading) and the 0.05% and 99.95% (thin lines) percentiles of the Monte Carlo distribution of randomly selected composite means. The percentiles depend on the size of random samples and are shown in blue and orange for the "weak" and "strong" composites, respectively.

of the hindcast, both reanalysis and the ensemble simulations exhibit a pronounced peak that is three times larger than the climatological average. Capturing this peak is crucial for predicting the onset of a SSW. Around Feb 20, reanalysis shows a second peak in planetary wave flux, which both hindcasts fail to predict and which likely contributes to the forecast error in the stratosphere. After the initial peak, the ensemble mean eddy-heat flux shows little difference from the climatological average (Fig. 3d), indicating limited predictability. In particular, there is no significant difference in heat flux between the model configurations with high and low vertical resolution. Therefore, we conclude that planetary waves do not play the predominant role in maintaining positive temperature anomalies in the later stage of the SSW and explaining the ensemble-mean temperature difference between the different model configurations. This leads to the hypothesis that the improvement in the temperature bias is rather due to the sensitivity of gravity waves to the vertical resolution. Hence, the remainder of this study focuses on the representation of gravity wave forcing in the model hindcasts.

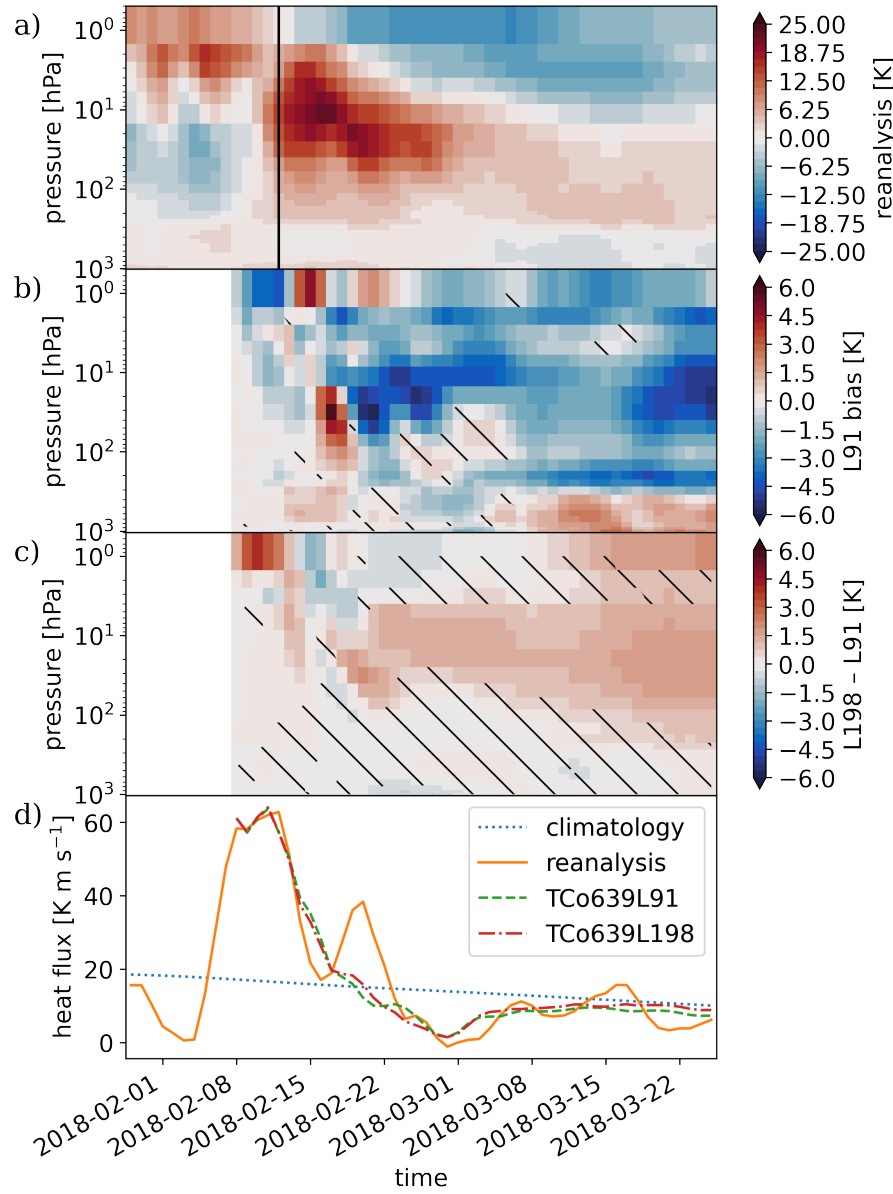

**Figure 3.** Polar cap temperature anomalies averaged between 60 °N and 90 °N during the 2018 SSW event. The central date of the event is marked by the reversal of zonal mean westerlies at 10 hPa and 60 °N indicated by the vertical line. (a) The development of temperature anomalies [K] in reanalysis, (b) the ensemble mean bias of TCo639L91 hindcasts initialized on Feb 08 compared to reanalysis, and (c) the improvement of the TCo639L198 hindcast ensemble mean compared to the TCo639L91 hindcasts. Hatching indicates areas where ensemble-mean differences are not significantly different from zero at a 95% confidence level estimated by a one-sample (b) or two-sample t-test (c). Panel d) shows the ensemble-mean meridional eddy heat flux by zonal wavenumbers 1 to 3 at 100 hPa averaged between 40 °N and 75 °N as a measure of the planetary wave flux in the lower stratosphere for climatology (dashed blue), the event in reanalysis (orange), and the two model configurations (green and red).





## 4 Stratospheric drag by gravity waves with small vertical wavelengths

Having established a sensitivity of the large-scale flow measured in terms of polar cap temperature anomalies to changing vertical model resolution, we are now looking for a mechanism to explain this sensitivity. Small-scale gravity waves can affect
the large-scale flow via non-linear processes such as wave breaking. The force on the zonal-mean flow exerted by the resolved part of the gravity wave spectrum can be diagnosed from the velocity output of the model in the form of the vertical eddy-momentum flux convergence (see Section 2). The force exerted by subgrid-scale gravity is output separately as accumulated momentum tendencies for orographic and non-orographic waves. The sum of those three forces is shown in Figure 4 where positive values mean an eastward acceleration of the zonal wind.

For an undisturbed vortex, gravity waves propagate into the mesosphere before they dissipate due to the reduced density (not shown). The disrupted vortex during the SSW, on the other hand, is further decelerated by gravity waves that dissipate in the mid-stratosphere below 5 hPa (Fig. 4). Above the center of upper-stratospheric easterlies during the major warming, the momentum tendency has positive values of more than $2 \, \mathrm{m \, s^{-1} \, day^{-1}}$, which corresponds to an acceleration of the disrupted vortex in the lower mesosphere. After the decay of upper-stratospheric easterlies around Feb 28, the mesosphere slowly returns
to values of about $-1 \, \mathrm{m \, s^{-1} \, day^{-1}}$ indicative of gravity waves with zero or negative phase speed reaching the mesosphere again. As noted in Section 1, the existence of a critical layer near the zero-wind line during SSWs has been reported by observational and modeling studies (e.g. Wang and Alexander, 2009; Yamashita et al., 2010). In the present hindcasts, however, westward drag in the mid-stratosphere is sustained after the return of the vortex to weak westerly. Thus, we conclude that the critical layer is located between the $0 \, \mathrm{m \, s^{-1}}$ and $10 \, \mathrm{m \, s^{-1}}$ isotachs indicated by solid black contours in Figure 4. This contrast to
earlier findings mentioned above is indicative of a broader spectrum of gravity waves than just stationary mountain waves.

As explained in Section 1 gravity waves can be expected to be sensitive to vertical resolution. To quantify this sensitivity, we investigate the difference in ensemble-mean gravity wave drag between the model configurations with high and low vertical resolution (Fig. 5). To verify that the expected change in resolved gravity wave drag with higher resolution is not compensated for by a change in parameterized momentum tendencies, the wave drag difference is split into its three components. The decel-
eration of the upper-stratospheric flow by resolved gravity wave drag in the simulations with higher resolution is continuously stronger by up to $0.2 \, \mathrm{m \, s^{-1} \, day^{-1}}$, which is consistent with increased temperature anomalies in the mid-stratosphere compared the configuration with low resolution (Fig. 5a). This difference in resolved gravity wave drag is not offset by parameterized waves. On the contrary, non-orographic gravity wave drag adds to the enhanced deceleration of the zonal-mean flow in the upper stratosphere (Fig. 5b), while the orographic gravity wave drag shows a difference only in the lower stratosphere (Fig.5c).
Note that the combined difference in Figure 5 represents almost 100% of full gravity wave drag at $10 \, \mathrm{hPa}$ in the hindcasts with 198 vertical levels seen in Figure 4. With 91 vertical level, there is zero wave drag on pressure surfaces around $100 \, \mathrm{hPa}$ indicating that, with low vertical resolution, gravity waves do not reach that altitude. This becomes clearer in the discussion of Figure 7. Also note that these results for the SSW in February 2018 are confirmed by an analysis of additional SSW events, which show a very similar behaviour in terms of absolute values and the difference between model configurations (Fig. A3).
Sensitivity tests with a different set of horizontal and vertical resolution are presented in Appendix B.



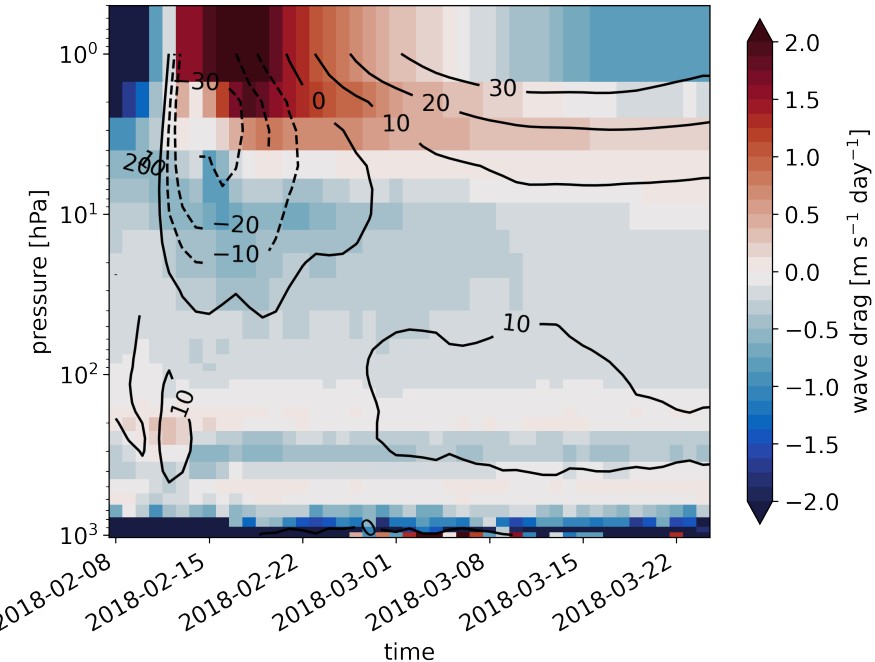

**Figure 4.** Ensemble-mean full gravity wave drag (resolved + orographic (parameterized) + non-orographic (parameterized)) horizontally averaged between 40 °N and 75 °N with contours of zonal-mean zonal wind $[\mathrm{m\,s^{-1}}]$ at 60 °N from the TCo639L198 hindcasts.

An important question when inferring whether gravity waves are responsible for the reduced temperature bias with increased vertical resolution is whether changes in gravity wave drag result directly from changes in resolution or whether they are a consequence of a modified background state. To assess this question, the diagnostics of Figure 5 have been repeated for the simulations where the mid- and upper-stratospheric zonal mean is nudged to reanalysis (Fig. 6). In these nudged simulations, the differences between the two model configurations inevitably result from increased vertical resolution since the stratospheric background state is unchanged. It is found that the sensitivity of resolved and parameterized wave drag does not depend on a difference in the zonal-mean background state, which supports the above hypothesis.

To better understand how the resolved gravity wave momentum flux depends on vertical resolution we aim to estimate the dominant vertical length scale resolved by the model. Therefore, we compute wavelet spectra of gravity wave potential energy from filtered temperature data as explained in Section 2. Note that gravity wave potential energy $E_{pot}$ is connected to the absolute value of the vertical flux of horizontal momentum $F$ according to the following identity (Ern et al., 2004)

$$F = \frac{k_h}{m} E_{pot}. \tag{5}$$

where $k_h$ and $m$ are the total horizontal and the vertical wavenumber. The model configuration with 198 vertical levels shows peak potential energy at a height of approximately 40 hPa for vertical wavelengths of about 6000 m, while the energy in



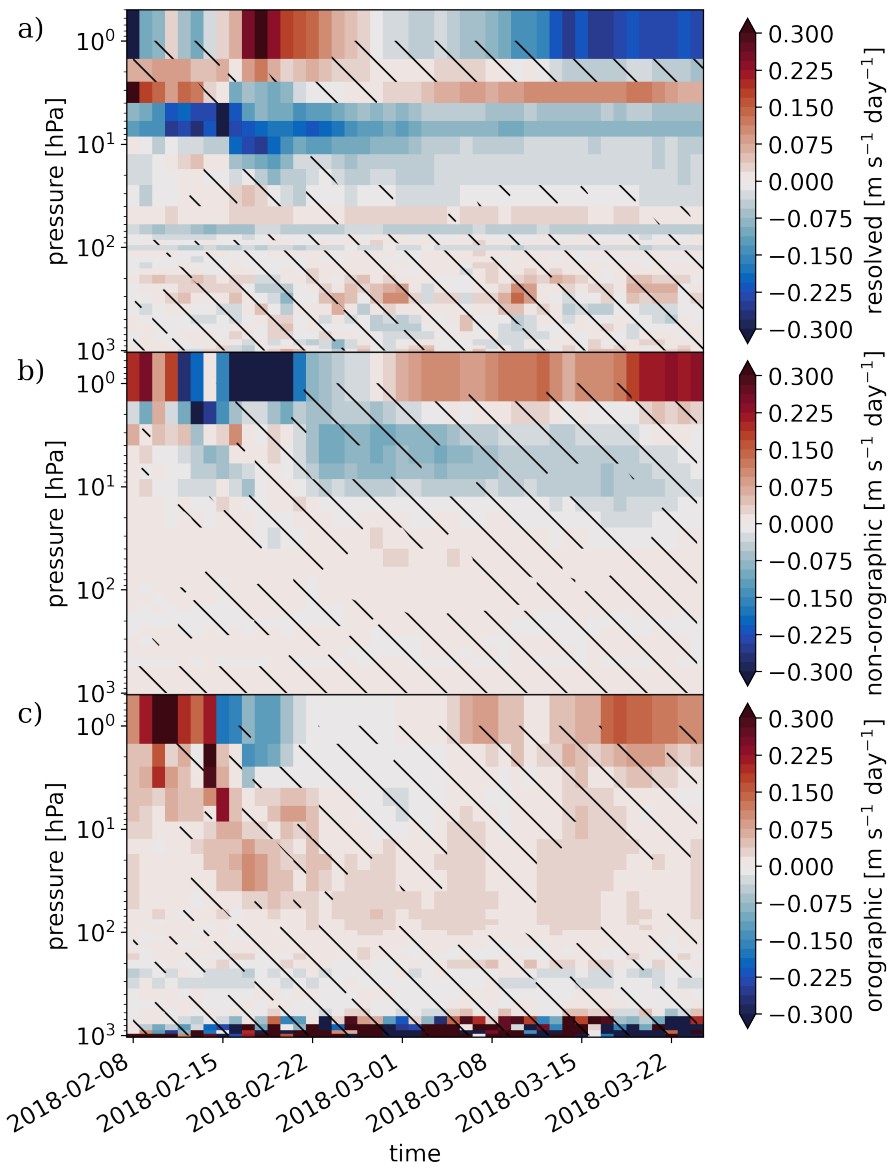

**Figure 5.** Difference of ensemble-mean gravity wave drag between the TCo639L198 and TCo639L91 free running hindcasts split into the (a) resolved, (b) non-orographic, and (c) orographic components. The wave drag is horizontally averaged between 40 °N and 75 °N and hatching indicates areas where the improvement with higher vertical resolution is not significantly different from zero at a 95% confidence level estimated by a two-sample t-test.

the model version with 91 vertical levels peaks already at 150 hPa and 8000 m wavelength (Fig. 7a, b). The overall scale of vertical wavelengths is roughly in line with observational estimates (e.g. Sato, 1994; Preusse et al., 2002). The global minimum of spectral power in the mesosphere above 40 km height can readily be explained by the gradually increasing

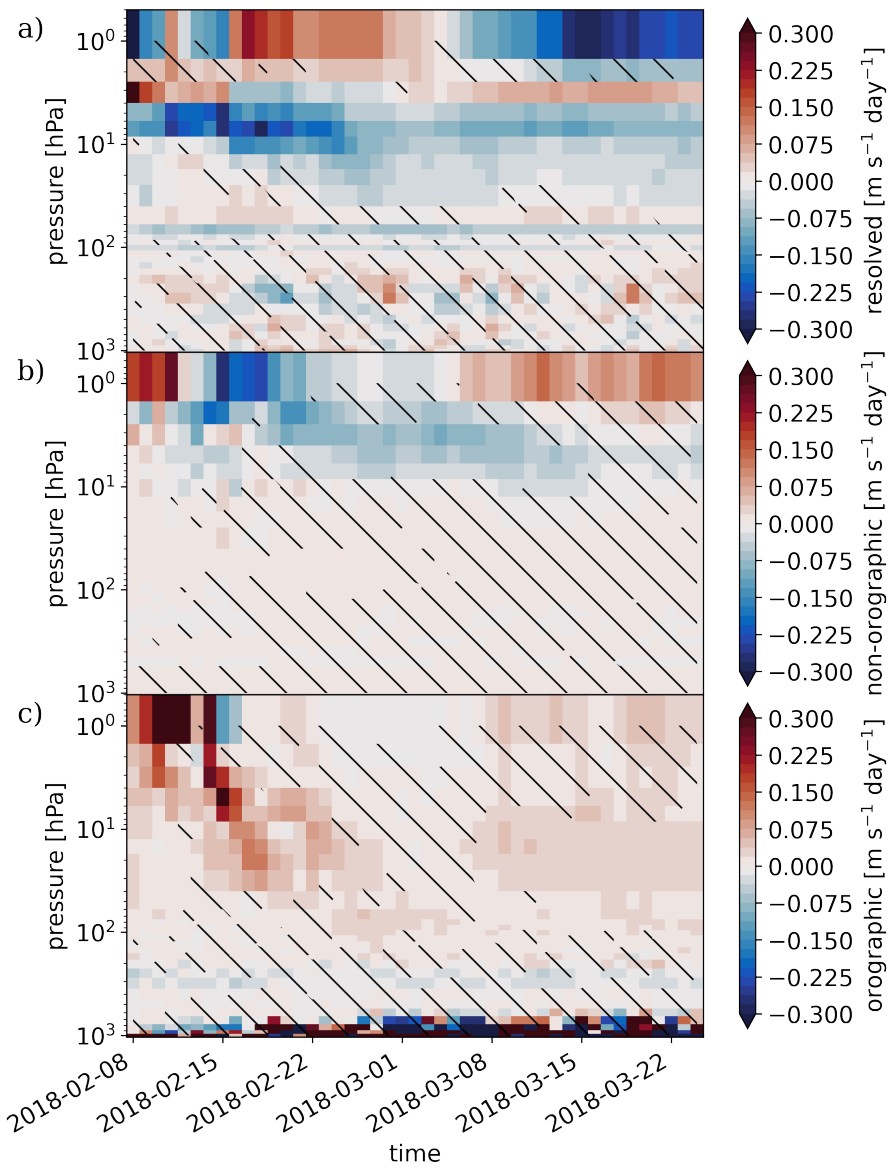

**Figure 6.** Same as Figure 5 but for the nudged simulations.

vertical grid spacing as we go higher up into the middle atmosphere (Fig. 1), the influence of the sponge layer at the model top, and the existence of a critical layer in the stratosphere. In the lower stratosphere there is a sign of a local minimum,

potentially indicative of a reflecting or refracting layer directly above the tropopause. A similar minimum can be observed in the gravity wave momentum flux at around 100 hPa (Fig. 4). However, the informative value of spectral power below 100 hPa in Figure 7 is reduced for two reasons. First, there are boundary effects caused by the convolution of the vertical column with





a Morlet wavelet marked by the cone of influence (hatched area). Second, the potential temperature anomalies are tapered by multiplication with a Hanning window prior to the wavelet transform to reduce spectral leakage at the boundaries which marks

a trade-off between removing artifacts and damping the signal. These limitations induced by the computation of the wavelet analysis are not easily overcome.

The difference of potential energy spectra between model configurations with high and low vertical resolution (Fig. 7c) shows a substantial increase of power with high resolution for vertical wavelengths of 4000 m and less in the mid- to upper stratosphere. The magnitude of this increase corresponds to more than 50% of the peak spectral energy in the configuration

with 91 model levels. Given the small spread between individual ensemble members of the same model configuration, the difference of potential energy spectra is highly significant almost everywhere. The wavelength for the strongest increase in power corresponds nicely to the difference in vertical grid spacing (Fig. 1), indicating that the difference in gravity wave drag seen in Figures 5, 6 does indeed result from gravity waves with small vertical wavelengths.

## 5   Conclusions

Previous studies (e.g. Karpechko et al., 2018; Lawrence et al., 2022) and a composite analysis in Section 3 reveal that the magnitude of stratospheric polar-cap temperature anomalies during an SSW is commonly underestimated in sub-seasonal prediction models. In order to better understand this underestimation, this study investigates the means of improving the prediction by increasing vertical model resolution. At a vertical resolution of 91 levels, the ensemble-mean error in targeted simulations of the SSW in 2018 reaches up to 4 K compared to reanalysis. By increasing vertical resolution from 91 to 198

vertical levels, the error is reduced by 50%, prolonging positive temperature anomalies in the polar stratosphere following SSW events. We here show that the most likely explanation for the strengthened and more persistent stratospheric warming signal is an improved representation of gravity wave dynamics with a higher number of vertical levels.

The present simulations are initialized four days prior to the onset of the SSW on February 12th, 2018. The model accurately predicts the strong flux of planetary waves into the stratosphere that caused the reversal of the prevalent westerlies. The asso-

ciated peak in planetary wave drag is one to two orders of magnitude larger than the persistent deceleration of the zonal-mean flow that is induced by resolved and parameterized gravity waves. However, the planetary wave flux in the weeks following the SSW has only limited predictability and no significant difference between the model configurations with different vertical resolution was found. In comparison to planetary waves, the ensemble spread of small-scale gravity waves is considerably smaller. The small variance in time and across ensemble members suggests that the sub-seasonal prediction skill is not significantly

hampered by a deterministic limit for predicting gravity wave source processes (e.g. Fritts and Alexander, 2003). The existence of a critical layer for near-stationary gravity waves is thus a good predictor for upper-stratospheric wave drag on sub-seasonal time scales. Gravity wave breaking near the critical layer constitutes a positive feedback between wave drag and the zonal-mean state of the vortex. To make full use of the critical layer as a predictor and to benefit from the aforementioned positive feedback, increased vertical resolution is necessary for an enhanced vertical gravity wave momentum flux. Specifically, it is



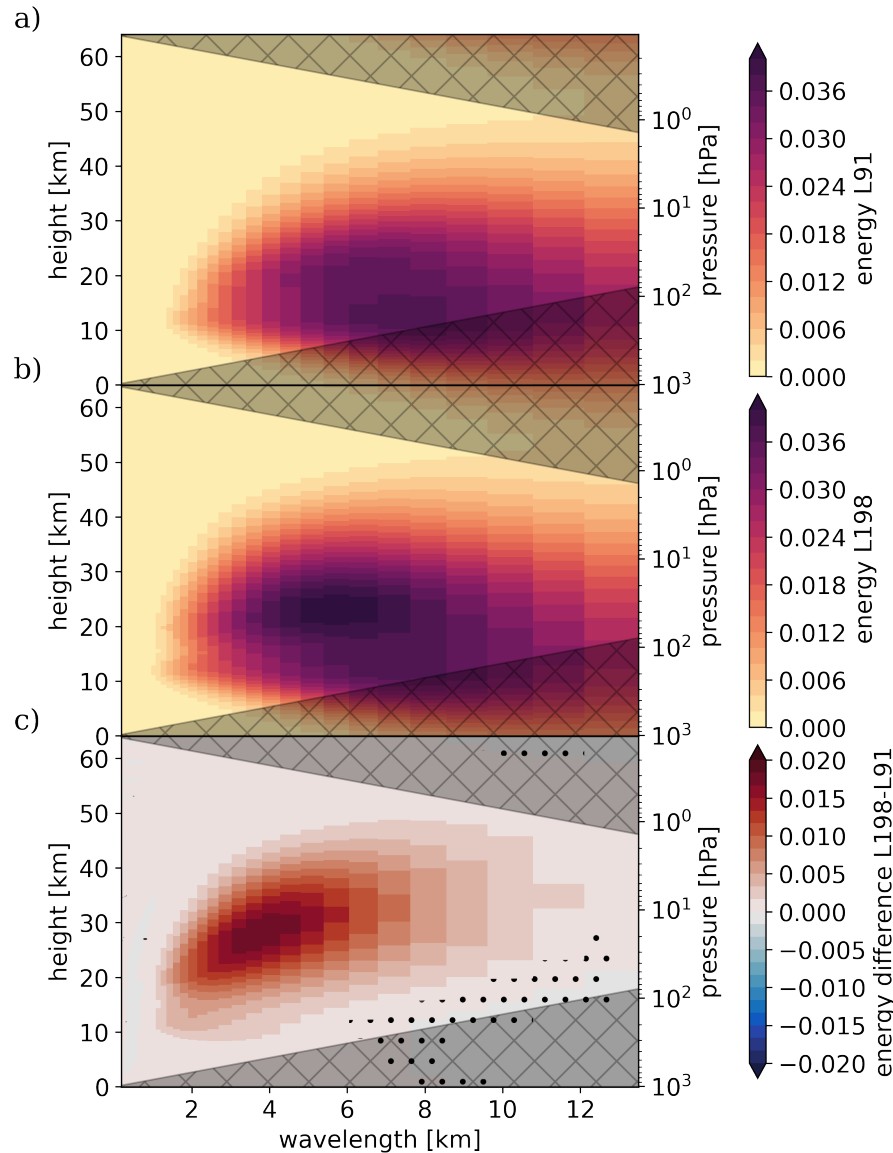

**Figure 7.** Ensemble-mean non-dimensional potential energy wavelet spectrum horizontally averaged between 40 °N and 75 °N for the period 2018-02-22 to 2018-03-22 in the TCo639L91 (a) and TCo639L198 (b) nudged simulations and the ensemble-mean difference TCo639L198 - TCo639L91 (c). The hatched area indicates the theoretical cone of influence and stippling in the lower panel indicates where ensemble-mean energies are not significantly different estimated by a parametric bootstrap.

found that a significant part of gravity wave potential energy corresponds to waves with small vertical wavelengths that are not resolved in the model configuration with only 91 vertical levels.





The need for a high vertical resolution to resolve gravity wave dynamics agrees well with previous modeling studies. Waite (2016) and Skamarock et al. (2019) find model convergence for a vertical grid spacing of 200 m at a horizontal resolution of 11.8 km and 15 km, respectively, which corresponds roughly to an aspect ratio of $f/N$. A similar conclusion is drawn in a

study about the sensitivity of the Integrated Forecasting System (IFS) to vertical resolution (Polichtchouk et al., 2019). While a vertical resolution set in accordance with the theoretical requirements to resolve stratified turbulence (see Section 1) performs well in the free troposphere, a correct representation of gravity wave breaking in the stratosphere requires a higher resolution than estimated for stratified turbulence (Cullen, 2017). Hence, stratospheric temperature biases in our model with a spectral resolution of TCo639 might benefit from an even further increase in vertical resolution compared to L198 (see Appendix B).

An open question that remains is to quantify the exact contribution of gravity wave drag to the observed reduction of the cold bias. There are two reasons that complicate such an estimate. First, the exact amount of wave drag required to maintain long-lasting positive temperature anomalies in the polar stratosphere against dissipation depends on the radiative damping time scale and the 'depth' of the warming (Hitchcock and Shepherd, 2013; Hitchcock et al., 2013). Second, the stationarity assumption inherent to the downward control principle (Haynes et al., 1991), which relates wave drag to adiabatic warming, is

violated by the transient nature of the experiments. Consequently, the sensitivity of polar cap temperature anomalies to vertical resolution might result from a reduced numerical error instead of adiabatic warming. While this study is concerned with the polar stratosphere during weak vortex conditions, the climatological cold bias in the absence of a critical layer is shown to be sensitive to vertical resolution, especially in the tropical stratosphere (Polichtchouk et al., 2019). That sensitivity is associated with discretization errors in vertical advection and an unphysical $2\Delta z$ mode in the temperature equation. But even though some

details of the mechanism remain unclear, the improvement of the stratospheric forecast with increased vertical resolution is evident.

**Appendix A**

This study presents results based on simulations of the SSW event in February 2018. The findings are robust and translate well to a greater number of SSW events as demonstrated by repeating the diagnostics for simulations of the events in January

2006 and February 2010. Figures A1 and A2 show the development of polar cap temperature anomaly for reanalysis and the hindcast ensemble mean as well as the ensemble-mean eddy heat flux at 100 hPa as a proxy for stratospheric planetary waves. The strong resemblance amongst different events in terms of gravity waves is shown by the ensemble-mean gravity wave drag in Figures A3a and A3b. The difference in resolved and parameterised wave drag between the model configurations with low and high vertical resolution is likewise comparable (Fig. A3c-h).

**Appendix B**

Operational sub-seasonal to seasonal forecasts issued by the ECMWF use 137 vertical levels and a horizontal resolution of TCo639 up to day 15 and a reduced horizontal resolution of TCo319 for the remainder of the forecast (ECMWF, 2022). The



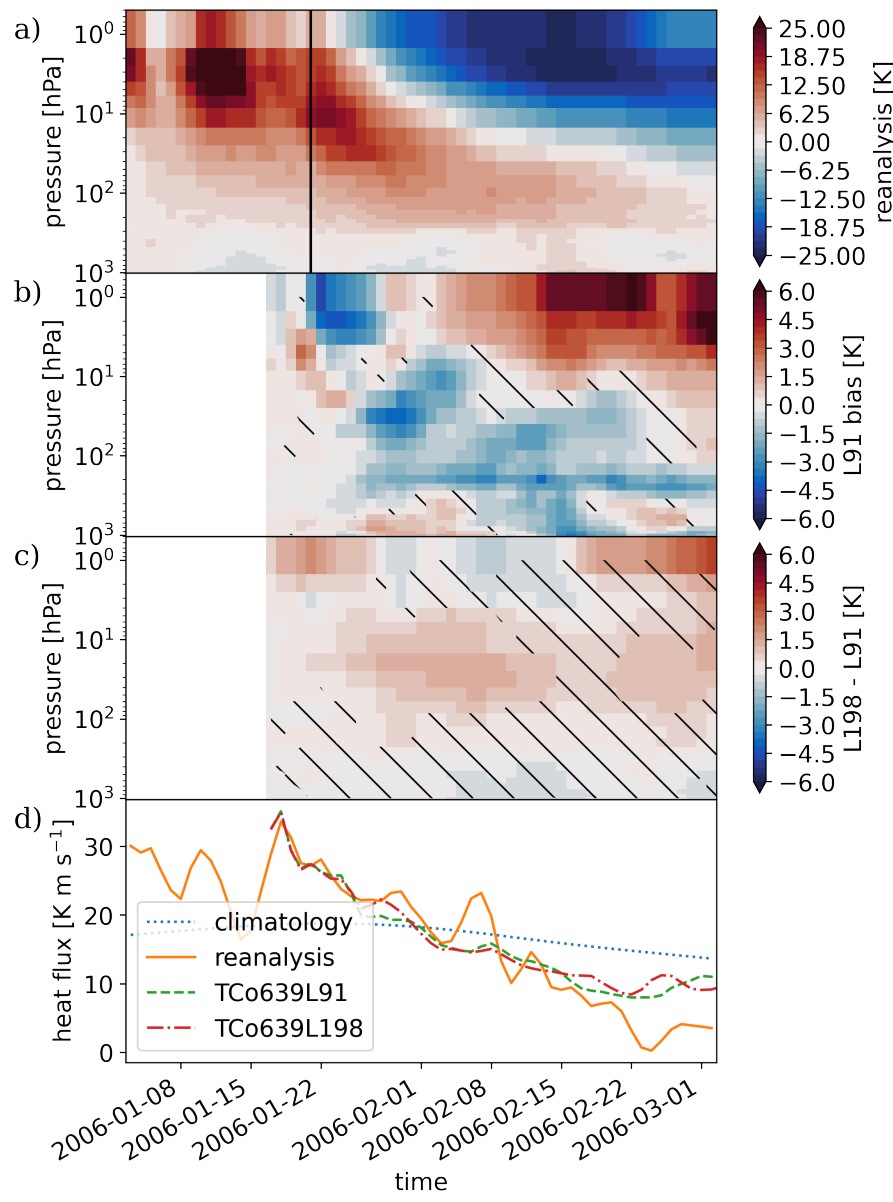

**Figure A1.** Same as Figure 3 but for the SSW event in 2006.

present study reveals an improved representation of stratospheric gravity waves and a reduction of the polar cap temperature bias in the TCo639L198 hindcasts compared to a model configuration with the same horizontal resolution but only 91 vertical 300 levels. To compare these results with the resolution of operational forecasts, we conducted sensitivity tests with a horizontal resolution of TCo319 and 137 vertical levels, respectively. They show that at lower horizontal resolution resolved gravity wave drag is reduced by 50% depending on altitude (Fig. B1). Furthermore, with TCo319 the improvement compared to 91



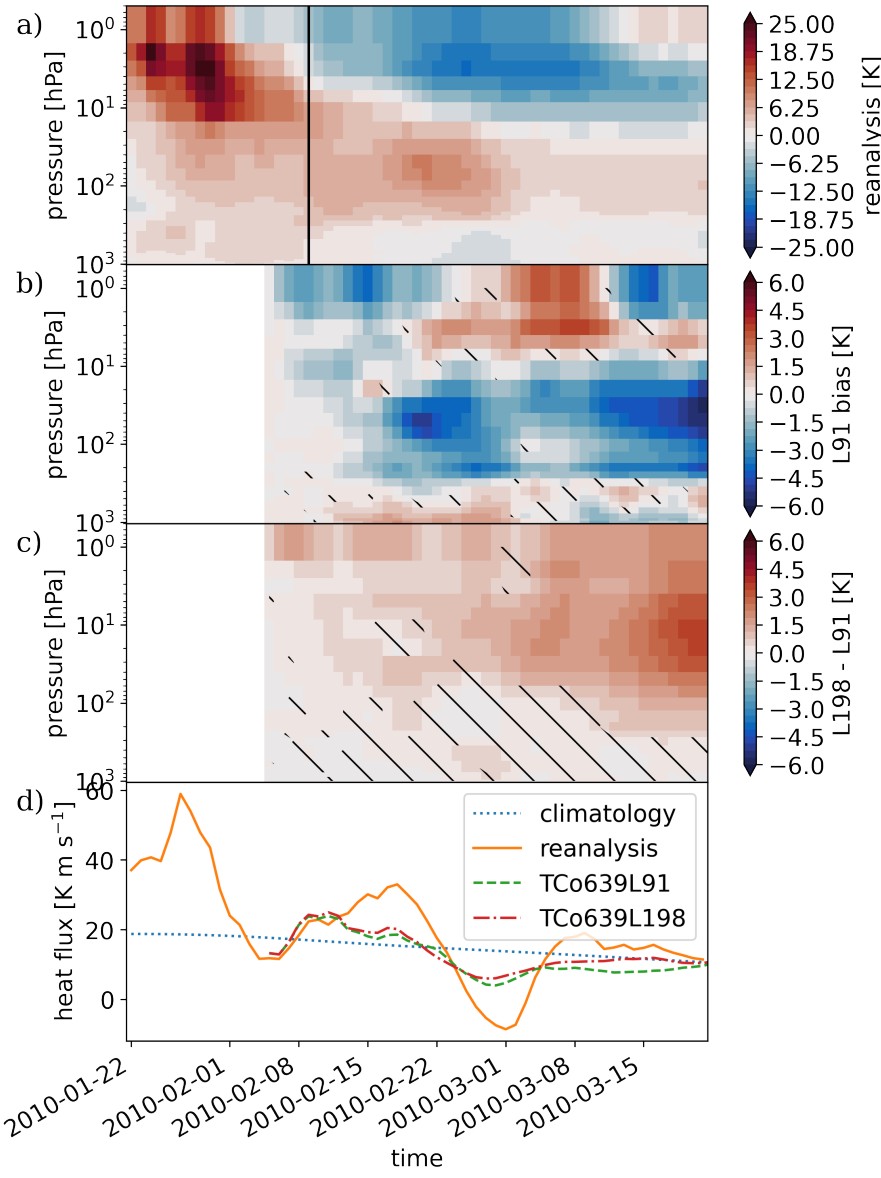

**Figure A2.** Same as Figure 3 but for the SSW event in 2010.

vertical levels is achieved to large degree already with 137 instead of 198 levels (Fig. B1). This is in contrast to the TCo639 simulations where increasing vertical resolution from L137 to L198 makes a significant difference (Fig. B2) especially in the upper stratosphere.




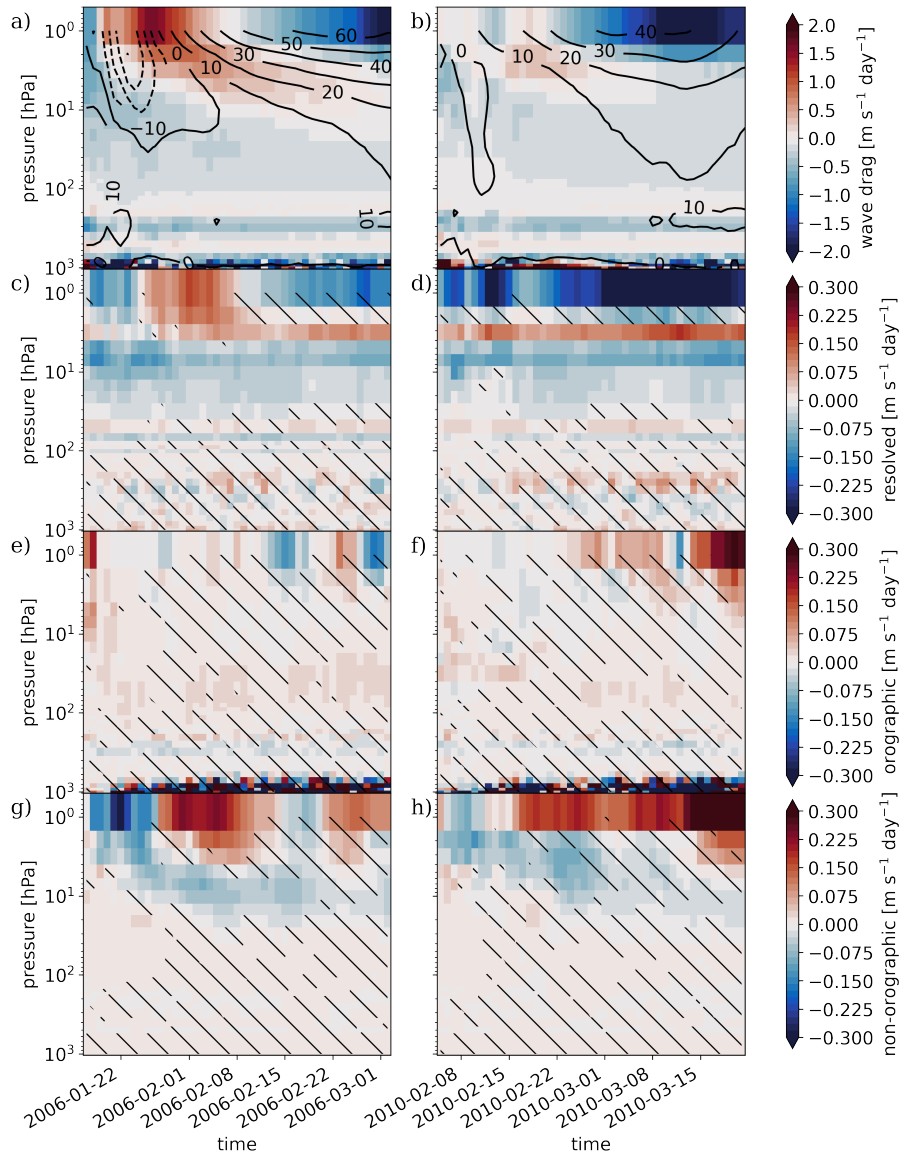

**Figure A3.** Ensemble-mean full gravity wave drag and zonal-mean zonal wind as in Figure B1 for the 2006 (a) and 2010 (b) SSW events. Differences in ensemble-mean gravity wave drag between the TCo639L198 and TCo639L91 free running hindcasts as in Figure 5 for 2006 (c, e, g) and 2010 (d, f, h) SSW events.

*Author contributions.* All authors jointly designed the study. I.P. performed the model runs and resolved gravity wave flux calculation. W.W. performed the data analysis, made the figures, and wrote the manuscript. All authors contributed to interpreting the results and editing the manuscript.



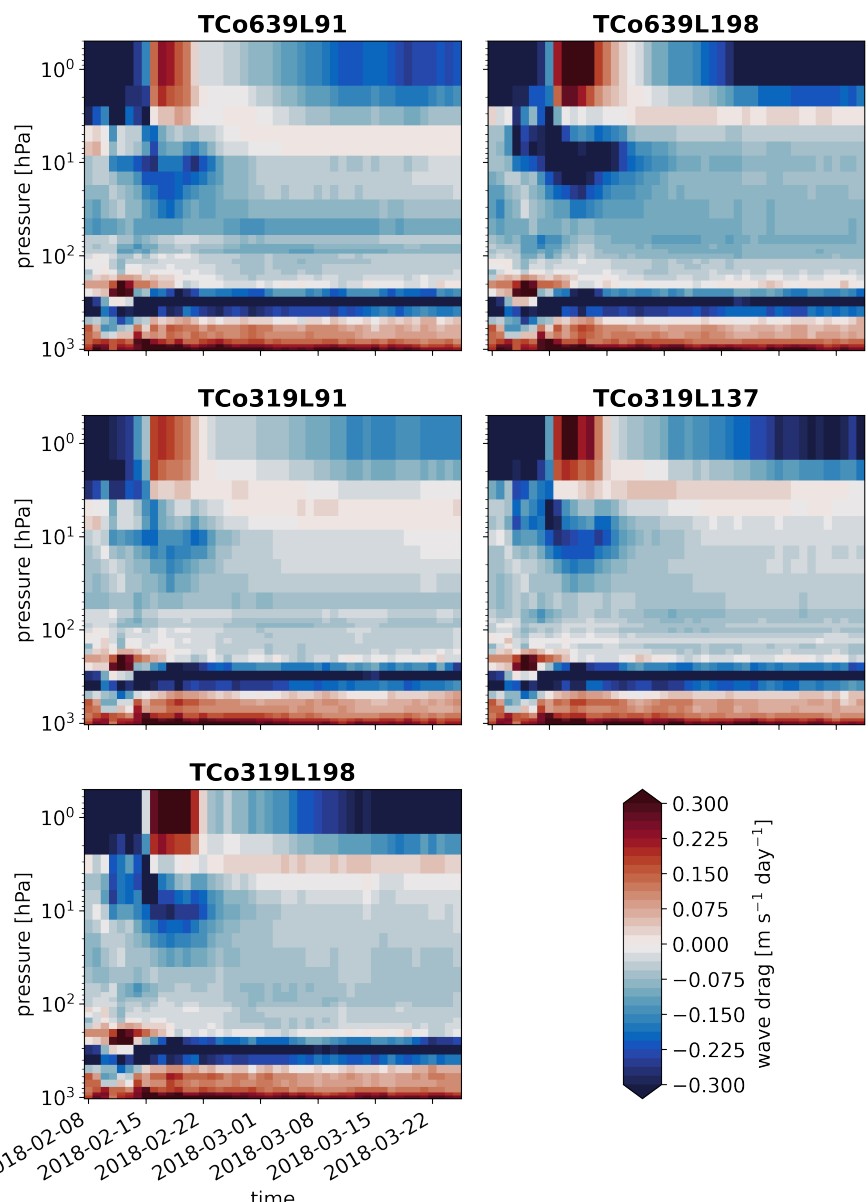

**Figure B1.** Ensemble-mean resolved gravity wave drag horizontally averaged between 45°N and 70°N for different model configurations initialized on February 08, 2018.

**Figure B2.** Differences of the TCo639L198 hindcast ensemble-mean polar cap temperature anomalies during the 2006, the 2010, and the 2018 SSW compared to the TCo639L137 hindcasts.



*Competing interests.* The authors declare no competing interests.

*Acknowledgements.* We thank Richard J. Greatbatch for insightful discussions and Tim Stockdale for providing the zonal-mean nudging code and for providing the 198L vertical resolution configuration. This project has received funding from the European Research Council (ERC) under the European Union's Horizon 2020 research and innovation programme (grant agreement No. 847456) and from the Swiss National Science Foundation through project PP00P2_198896.



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
