# Peer review of "Increased vertical resolution in the stratosphere reveals role of gravity waves after sudden stratospheric warmings"

_Weather and Climate Dynamics, 2022_

## Referee Comment (RC2)

Sincere apologies for a slower than hoped response with this review.

This is a nicely done study, with a set of results that show a clear sensitivity of gravity waves and drag to vertical resolution in the IFS for SSW events. The results also clearly show:
  (1) That the higher vertical resolution leads to more stratospheric drag by resolved gravity waves in the weakened wind conditions during and following SSW events.
  (2) That higher vertical resolution becomes more important in IFS with finer horizontal resolution.
  (3) That the increased vertical resolution resolves more short vertical wavelength gravity waves in the stratosphere.
This is a fine and useful study utilizing state-of-the-art and computationally intensive high-resolution simulations that are not widely available or analyzed in this way. It will certainly make a fine addition to the literature and makes clear some important new points.

There is one smaller but I think important issue with one specific conclusion drawn that is not supported by the presented results. To go one step further, the results may be explained without this unsupported conclusion. Please see major comment (1). I suggest here a few changes and a suggested alternate for this minor conclusion as a way for the authors to proceed without any new calculations, but alternatively if a bit more work is done to support the current conclusion, that could be another approach.

Major comment (2) itemizes a few places where the current text wording communicates some misunderstanding about gravity wave breaking and critical levels that could be corrected with only minor rephrasing. These few sentences are also relevant to the misstated conclusion in (1).

Major comments:

(1) L263: "To make full use of the critical layer as a predictor and to benefit from the aforementioned positive feedback, increased vertical resolution is necessary for an enhanced vertical gravity wave momentum flux. Specifically, it is found that a significant part of the gravity wave potential energy corresponds to waves with small vertical wavelengths that are not resolved in the model configuration with only 91 vertical levels."
While the second sentence is supported by Figure 7, nowhere is it shown that there is "an enhanced vertical gravity wave momentum flux." Figure 5 shows enhanced drag at higher vertical resolution. Figure 7 clearly shows enhanced potential energy at shorter vertical wavelengths, and it appears that the enhancement appears at longer vertical wavelengths at higher altitudes, possibly in concert with the changes in resolution with height (Fig. 1). At face value, this Fig. 7 seems to show that improving vertical resolution prevents artificial dissipation at too-coarse vertical scales, and that the improved resolution allows waves to propagate a bit closer to critical levels, delaying dissipation until higher altitude. This delayed dissipation effect alone can increase drag without any increase in momentum flux. The 1/density factor in the drag equation will give larger drag if the same wave momentum flux is dissipated at a higher altitude. (See for example Vincent and Alexander (2020) and text surrounding their equation (2) in the context of the QBO shear zones.) To claim instead that the momentum flux in the stratosphere is higher at higher vertical resolution, the flux profiles would need to be computed and compared. A simpler approach would be to discuss the enhanced waves in the high vertical resolution simulation in terms of this delay in artificial dissipation at too-short vertical scales. Perhaps this was what is intended, and if so please find several suggested changes to the text to clarify things below in (2), which I hope may improve the paper for readers.

(2) Please find here several suggested changes (by line number) to improve the descriptions related to gravity wave breaking/dissipation and critical levels. Wave dissipation/breaking in a model like

this will always occur somewhere significantly below a critical level. Using the word "layer" instead of "level" doesn't really solve the problem.

**L29-31**: "These waves propagate via the stratosphere into the mesosphere, where their amplitudes grow until they break. However, depending on their phase speed and the background wind, gravity waves can encounter a critical layer, where vertical length scale shrinks to zero, and deposit their momentum already at lower altitudes in the stratosphere."
It would be best to begin L29 with "*Commonly*, these waves propagate…" While it is a general tendency for gravity waves in models like IFS to have their biggest impacts in the mesosphere, real atmospheric gravity waves have a wide distribution of amplitudes, and many dissipate/break in the stratosphere (e.g. de la Camara and Lott, 2014; 2015). Suggested change is to add the word "Commonly," at the beginning of this sentence. The second sentence might be better phrased as e.g., "…depending on their amplitude, phase speed, and the background wind, gravity waves would encounter a critical level (where vertical wavelength shrinks to zero), and break somewhere below that level, depositing more momentum at lower altitudes in the stratosphere."

**L190-195**: Wind changes and associated changes in wave critical levels were discussed by these earlier references more generally than implied here. Wind contours can show limits where any gravity waves with phase speeds within certain wind speed ranges will certainly not be able to propagate through that level, but the phase speed range of breaking waves will not be equal to the wind speed range because wave breaking/dissipation will generally occur somewhere below the wave critical level (particularly for models, including IFS, which has good but still limited vertical resolution). Suggested small wording changes:
      L190 change "zero or negative phase speed" to "low negative phase speeds"
      L193 change "critical layer" to "breaking layer" or "dissipation layer"
      L195 change "is indicative" to "may be indicative"

**L206-207**: "With 91 vertical level, there is zero wave drag on pressure surfaces around 100 hPa indicating that, with low vertical resolution, gravity waves do not reach that altitude. This becomes clearer in the discussion of Figure 7." Zero drag at 100hPa indicates no dissipation/breaking at that level, but waves with large momentum flux could pass through without dissipation and no drag. To instead say some waves are missing at 100hPa, would require comparison of the momentum fluxes. So it would only be accurate if you change "do not reach" to "do not break at". I might suggest deleting both sentences since the first statement does not become clearer after Fig.7. (See above.)

**L266**: Suggest some additional clarification after L266 to mention that the enhanced scales get longer at higher altitudes and may be related to the profile of vertical resolution shown in Fig.1, as mentioned above. e.g. enhanced vertical wavelengths $\sim 4-10\Delta z$? Again, restating an earlier comment, I do not see any clear evidence in Fig. 7 that there are enhancements in wave fluxes crossing the tropopause.

Minor comments:

L77: Typo, e.g missing word "mitigated by" or "mitigated in"
L115: "normal mode". These aren't normal modes. Perhaps change to "wave mode" or "mode"?
Figure 5 caption, add word "parameterized" after (b) and (c) for clarity.

---

## Author Comment (AC1)

**Final author comments**

We would like to thank Anonymous Referee 1 and M. Joan Alexander for their insightful and constructive review of our manuscript and for the time to review our work. Their comments and suggestions improve the paper for readers substantially. Please find our detailed responses to their respective comments below. All line indications apart from those in quotation marks refer to the new (annotated) version of the manuscript. All changes in the annotated manuscript compared to the original version are indicated in **bold**.

**Main comment concerning gravity wave momentum fluxes**

In response to general point 3 by Referee 1 and major comment 1 by M. Joan Alexander, we have computed profiles of gravity wave momentum flux and found no sensitivity to vertical resolution at the tropopause level. The gravity wave drag D analysed in our manuscript is computed in pressure coordinates where the 1/density factor mentioned by M. Joan Alexander is not immediately visible. In log-pressure coordinates though, the drag can be expressed as follows:

$$D = -\frac{\partial}{\partial p} \left( \overline{u'\omega'} \right) = -\frac{1}{\rho_0} \frac{\partial}{\partial z} \left( \rho_0 \overline{u'w'} \right) \tag{1}$$

The ensemble-mean profile of  $\rho_0 \overline{u'w'}$  at high vertical resolution (Fig. 1, left panel) exhibits a strong upward flux of easterly momentum in the troposphere and gradually diminishing values from the lower to the mid-stratosphere. The ensemble-mean difference between the two model configurations (Fig. 1, right panel) exhibits no statistical significance below 70 hPa indicating that tropospheric gravity wave sources are not significantly sensitive to the different grid configurations. It is only in the mid-stratosphere and above, that the propagation of gravity waves becomes sensitive to vertical resolution due to the gradually increasing grid spacing. With low vertical resolution, gravity waves dissipate at a lower altitude compared to the model configuration with high resolution, and thereby exert a reduced drag on the zonal-mean flow, as explained in the comments by M. Joan Alexander.

Figure 1: Left panel: ensemble-mean gravity wave momentum flux  $\rho_0 \overline{u'w'}$  averaged between 45°N and 70°N with contours of zonal-mean zonal wind [m s-1] at 60°N from the TCo639L198 hindcasts. Right panel: difference of ensemble-mean gravity wave momentum flux between the TCo639L198 and TCo639L91 hindcasts averaged between 45°N and 70°N and hatched where not significantly different from zero at a 95% confidence level.

**Anonymous Referee 1:**

**General points**

"I like the concept of using the 2018/19 SSW as primary case, but also including additional analyses (2 more SSWs and S2S diagnostics) to ensure the robustness of your results."

**Answer:** We appreciate that the reviewer finds our approach of a three-event case study convincing. Please note that the SSW in February 2018 serves as the primary case. We have made this more explicit in the manuscript (see response to specific point 4).

"I am wondering a bit about the 198 levels; do you think the gravity waves are sufficiently resolved at this resolution or should the observed bias decrease further with more levels? If the drag has converged, is there any argument why this should be a suitable resolution (eg would this be generally consistent with estimations based on Eq. 3 or so)? Of course, this is generally discussed in the paper (eg Figs. B1/B2 and Sec. 5) but can you give any estimate of at what resolution you would expect convergence?"

Answer: Thank you for this question. As noted by the reviewer, the level of vertical resolution required to achieve model convergence is discussed in the third paragraph of Section 5. Waite (2016) and Skamarock et al. (2019) find model convergence for an aspect ratio of vertical to horizontal resolution of about f/N. In the free troposphere above the planetary boundary layer and the lower stratosphere, the 200m grid spacing of the L198 vertical grid is, probably, appropriate for the wave spectrum resolved by the horizontal resolution of TCo639. To make full use of the stratospheric gravity wave drag during weak vortex conditions, this grid spacing should, ideally, be maintained up to the stratopause. A realistic simulation of the gravity wave breaking itself, where the vertical

wave length approaches zero, would require much higher resolution that is not currently feasible.

"Can you say something about differences in tropospheric gravity wave forcing? I think your results are generally convincing (especially since you find robust signals for 3 different SSWs), but I guess generally differences/biases in wave drag could in principle result from changes/biases in wave forcing? Did you look at corresponding signals/correlations for your 3 SSWs or maybe within your S2S dataset?"

**Answer:** Thank you for this question. Please see also the response to major comment 1 by M. Joan Alexander and the description of Figure 1 above. Specifically, we see no resolution sensitivity of the gravity wave momentum flux in the lowermost stratosphere.

**Specific points**

"L58-59: Maybe specifically include the estimation "scale height/Rossby radius" in equation 1?" **Answer:** Thank you for this suggestion. We have changed equation 1 accordingly.

"L71: model experiments"

**Answer:** Thank you for spotting this typo, which is corrected in line 70.

"L75: This might be a personal preference, but I was never a fan of the word "confirm" in a scientific context. Maybe just say that you find a corresponding bias in S2S data?" **Answer:** Thank you for this suggestion. We have changed the phrasing in line 74 to a quantification of the bias.

"L81: It is clear from the plots later, but maybe clarify here specifically if you mean 2017/18 or 2018/19 since both winters had a SSW."

Answer: Thank you. It is made more clear in line 80.

"Fig. 2: Just for clarification: is the initial bias a result of the way you construct your climatology from different hindcasts?"

**Answer:** Thank you for this question. Figure 2 does not involve any computation of a model climatology. The bias is computed directly as the difference of polar cap temperature between hindcast and reanalysis. The exact reason for the initial bias eludes us, but we expect that it is connected to the procedure for model initialization or the production of model output.

"L193-194: Most wave theories are based on linear and other crude assumptions and the applicability of inferred quantitative values to the real atmosphere is always a bit limited. I am not sure if it is useful to state a specific quantitative range for your critical wind lines here since you

don't actually use it anywhere else in the paper."

**Answer:** Thank you for this comment. We have removed the corresponding passage in line 194 from the manuscript.

"Fig 5 and some others: I think the figure would be much easier to read if you only include one colour bar and distinguish the different panels with panel headings."

**Answer:** Thank you for this suggestion. We have changed the layout of Figures 3, 4, 5, 6, 7, A1, A2, A3, and B2 in order to share colour bars among multiple panels.

**M. Joan Alexander:**

**Major comments**

"[...] nowhere is it shown that there is "an enhanced vertical gravity wave momentum flux." Figure 5 shows enhanced drag at higher vertical resolution. Figure 7 clearly shows enhanced potential energy at shorter vertical wavelengths, and it appears that the enhancement appears at longer vertical wavelengths at higher altitudes, possibly in concert with the changes in resolution with height (Fig. 1). At face value, this Fig. 7 seems to show that improving vertical resolution prevents artificial dissipation at too-coarse vertical scales, and that the improved resolution allows waves to propagate a bit closer to critical levels, delaying dissipation until higher altitude. This delayed dissipation effect alone can increase drag without any increase in momentum flux. The 1/density factor in the drag equation will give larger drag if the same wave momentum flux is dissipated at a higher altitude. [...]"

**Answer:** Thank you for this comment. We have added plots of the resolved gravity wave momentum flux in Figures 4b and 5d. These are referenced in lines 190 f. and 205 ff. The term gravity wave flux is replaced by gravity wave drag in lines 95 and 231. The sentence in line 264 has been rephrased emphasizing the wave drag instead of the wave momentum flux.

"Please find here several suggested changes (by line number) to improve the descriptions related to gravity wave breaking/dissipation and critical levels. Wave dissipation/breaking in a model like this will always occur somewhere significantly below a critical level. Using the word "layer" instead of "level" doesn't really solve the problem."

**Answer:** Thank you for these suggestions.

- In lines 29-33, we have rephrased three sentences to make clear that gravity waves break before they reach the theoretical critical layer where the vertical wavelength would reach zero. The new formulation is: "These waves commonly propagate via the stratosphere into the mesosphere, where their amplitudes grow until the waves break. Depending on the phase speed of the waves and the velocity of the background wind, one can define a critical layer where the intrinsic frequency of the waves  $\hat{\omega}$  would approach the inertial frequency f and the vertical wavelength would approach zero (e.g. Fritts and Alexander, 2003). If such a critical layer is present, gravity waves will break somewhere below that level and deposit more momentum already in the stratosphere."

- The wording has been changed in lines 189, 192, and 194. Two sentences in line 194 have been removed from the manuscript. Two more sentences in line 205 have been removed from the manuscript. A clarifying sentence is added in line 266.

**Minor comments**

"L77: Typo, e.g missing word "mitigated by" or "mitigated in"" Answer: Thank you for spotting this typo. It is corrected in line 76.

"L115: "normal mode". These aren't normal modes. Perhaps change to "wave mode" or "mode"?" **Answer:** Thank you for this suggestion. It is changed to "wave mode" in line 114.

"Figure 5 caption, add word "parameterized" after (b) and (c) for clarity." Answer: Thank you for this suggestion. We have changed the caption of Figure 5 accordingly.

**Modifications of the manuscript**

In addition to changing the manuscript in response to the referees' comments, we have made the following modifications to the manuscript:

- A small change to one sentence in the abstract at line 4.
- A more clear formulation in lines 48-53 with additional reference to a recent publication by Wu et al. (2022).
- A small change to the sentence in line 97 with additional reference to a recent publication by Polichtchouk et al. (2022).
- A more clear formulation in lines 161-162.
- A corrected typo in the caption of Figures 3, 4, 5, and 7 about the latitudinal range of wave drag averaging.
- Small editorial corrections throughout the manuscript.